# Evaluation of Inference Performance of Deep Learning Models for Real-Time Weed Detection in an Embedded Computer

**DOI:** 10.3390/s24020514

**Published:** 2024-01-14

**Authors:** Canicius Mwitta, Glen C. Rains, Eric Prostko

**Affiliations:** 1College of Engineering, University of Georgia, Athens, GA 30602, USA; 2Department of Entomology, University of Georgia, Tifton, GA 31793, USA; 3Department of Crop and Soil Sciences, University of Georgia, Tifton, GA 31793, USA; eprostko@uga.edu

**Keywords:** weed detection, precision weeding, deep learning weed detection, weed detection inference in embedded computer

## Abstract

The knowledge that precision weed control in agricultural fields can reduce waste and increase productivity has led to research into autonomous machines capable of detecting and removing weeds in real time. One of the driving factors for weed detection is to develop alternatives to herbicides, which are becoming less effective as weed species develop resistance. Advances in deep learning technology have significantly improved the robustness of weed detection tasks. However, deep learning algorithms often require extensive computational resources, typically found in powerful computers that are not suitable for deployment in robotic platforms. Most ground rovers and UAVs utilize embedded computers that are portable but limited in performance. This necessitates research into deep learning models that are computationally lightweight enough to function in embedded computers for real-time applications while still maintaining a base level of detection accuracy. This paper evaluates the weed detection performance of three real-time-capable deep learning models, YOLOv4, EfficientDet, and CenterNet, when run on a deep-learning-enabled embedded computer, an Nvidia Jetson Xavier AGX. We tested the accuracy of the models in detecting 13 different species of weeds and assesses their real-time viability through their inference speeds on an embedded computer compared to a powerful deep learning PC. The results showed that YOLOv4 performed better than the other models, achieving an average inference speed of 80 ms per image and 14 frames per second on a video when run on an imbedded computer, while maintaining a mean average precision of 93.4% at a 50% IoU threshold. Furthermore, recognizing that some real-world applications may require even greater speed, and that the detection program would not be the only task running on the embedded computer, a lightweight version of the YOLOv4 model, YOLOv4-tiny, was tested for improved performance in an embedded computer. YOLOv4-tiny impressively achieved an average inference speed of 24.5 ms per image and 52 frames per second, albeit with a slightly reduced mean average precision of 89% at a 50% IoU threshold, making it an ideal choice for real-time weed detection.

## 1. Introduction

Invasive weeds in agricultural fields provide competition for crucial resources to crops. For most crops, weeds cause higher losses in production than pathogens and animal pests [1], underscoring the importance of control. Weed control has proven to be a significant challenge. Herbicides have been the go-to method of controlling weeds for decades [2,3], in addition to other common solutions like mechanical weeding [4,5,6] and even hand-picking.

The evolution of herbicide-resistant weed populations threatens agricultural productivity [7,8]. In addition to this, herbicides and other conventional methods of weed control such as mechanical techniques are labor-intensive and expensive [9]. Technology provides an opportunity to increase efficiency in control and reduce costs. Weed control solutions that automate the entire process or part of the process such as automatic sprayers [9,10,11] and precision mechanical weed controllers [12,13] have been researched and implemented.

Precision weed control methods demand knowledge of the types and location of weeds in the field; therefore, weed detection solutions are essential for this task. Research into weed detection technologies has resulted in various solutions that have proven valuable for precision weed control. Some solutions have used infrared spectroscopy [14], fluorescence [15], or computer vision [16,17,18]. The availability of low-cost high-resolution cameras and advances in computing hardware have sparked interest in computer vision solutions. Some scholars have used a combination of simple image-processing techniques that utilize the extraction of features like color, shape, or texture and machine learning algorithms like support vector machines or random forest to identify weeds. For example, the authors of [16,17,18] used a combination of image feature extraction and support vector machines to discriminate weeds from crops. While these methods perform well in stable environments, they may not be robust in harsh outdoor conditions with changes in illumination, occlusions, and shadows. Progress in deep learning technologies has led to an increase in the use of convolutional neural networks (CNNs) for weed detection and classification, achieving impressive results. For instance, the authors of [19,20,21,22] used CNN frameworks to detect weeds in crop fields with a great level of stability and accuracy. Individual weed detection makes it possible to implement weed removal solutions that can precisely target individual weed species without interfering with other plants in the field. These solutions include methods like spot spraying [23], electricity [24], or lasers [25].

Researchers have compared the effectiveness and efficiency of different deep learning models in detecting different weed species, aiding informed decision making for field implementation. For example, ref. [26] compared two deep learning models, the single-shot detector (SSD) and Faster RCNN, according to their detection performance on UAV imagery and found Faster RCNN to be the superior model. In another study, the authors of [27] evaluated 35 deep learning models on 15 weed species, establishing a benchmark for weed identification. Most of the performance evaluations and comparisons for deep learning models have been conducted on powerful computers capable of handling the computational demands of deep learning; however, given our focus on robotic applications, many solutions require portable computers, which are often less powerful. For real-time applications in agricultural fields, robotic platforms such as ground rovers and UAVs usually use embedded computers that are not comparable to most powerful GPU-enabled computers used for deep learning tasks.

This paper compares the performance of three single-stage deep learning models that are lightweight enough for real-time applications: YOLOv4 [28], EfficientDet [29], and CenterNet [30]. The comparison focuses on the real-time detection of thirteen common species of weeds found in cotton and peanut fields. The comparisons were conducted in a deep learning computer with powerful GPUs (RTX 2080Ti, Nvidia, Santa Clara, CA, USA) and an embedded deep-learning-enabled computer (Nvidia Jetson Xavier AGX). These models were chosen due to their reputation as state-of-the-art object detection models for real-time applications. Detecting multiple species of weeds individually can help in making real-time decisions about how to remove the weed; for example, if a robotic platform is performing spot spraying and encounters an herbicide-resistant weed, an alternative method can be employed.

## 2. Materials and Methods

### 2.1. Data Collection

More than 5000 color or RGB (Red, Green, Blue) images of 13 different weed species—Palmer amaranth (*Amaranthus palmeri*), smallflower morningglory (*Jaquemontia tamnifolia*), sicklepod (*Senna obtusifolia*), crabgrass (*Digitaria* spp.), Florida beggarweed (*Desmodium tortuosum*), Florida pusley (*Richardia scabra*), pitted morningglory (*Ipomoea lcunos*), goosegrass (*Eleusine indica*), crowfoot grass (*Dactyloctenium aegyptium*), purple nutsedge (*Cyperus rotundus*), yellow nutsedge (*Cyperus esculentus*), ivyleaf morningglory (*Ipomoea hederacea*), and Texas panicum (*Urochloa texana*), seen in Figure 1—were collected from University of Georgia research fields near Ty Ty, GA (31.509730 N, 83.655880 W) and the University of Georgia Tifton campus, GA (31.473410° N, 83.530475° W) using smartphone cameras or hand-held digital cameras. Images were captured at early stages of weed growth (from 1 to 3 weeks) at different camera angles, under different weather conditions, and at different times of the day.

### 2.2. Data Labeling

More than 3500 images were labeled using an open-source annotation tool, LabelImg v1.8.6 (https://github.com/HumanSignal/labelImg, accessed on 10 February 2022). This tool allows for the drawing of boundaries around objects in images to identify them and creates records that indicate the object’s location in the image, as seen in Figure 2. Labeling was conducted in both PASCAL VOC [31] format for TensorFlow model training and YOLO [32] format for YOLO model training in darknet.

### 2.3. Train–Test Split

The labeled data were divided into a training set (60%) for training the models to learn the features, a validation set (20%) to validate the model’s precision and avoid overfitting, and a testing set (20%) for benchmarking, as shown in Figure 3.

### 2.4. Data Augmentation

Since deep learning models rely heavily on extensive data for improved accuracy and to prevent overfitting, any additional data are valuable. Data augmentation involves techniques that add slightly modified copies of the existing data to the training set to enhance the size and quality of training data [33].

The training data were augmented through techniques such as rotation, shearing, blurring, and cropping using an open-source image augmentation library, CLoDSA (https://github.com/joheras/CLoDSA, accessed on 2 May 2022). This increased the training set to more than 67,000 images.

### 2.5. Training

Training was conducted using transfer learning, a technique of transferring knowledge between different but related domains [34]. In deep learning, this is accomplished by reusing previously trained models for new problems to reduce training time and enhance the performance of targeted models. In training, the models take labeled images of different resolutions and then change the resolution to the required model input size.

#### 2.5.1. YOLOv4

YOLOv4 (You Only Look Once version 4) is a real-time object detection model developed as a continuation of previous YOLO versions to address their limitations. It is a single-stage object detection model trained to analyze the image only once and identify a subset of object classes. The YOLO network architecture is renowned for its speed in object detection, and YOLOv4 has prioritized real-time detection.

YOLOv4 training was conducted under the darknet environment [35], which is an open-source neural network framework that supports object detection and image classification tasks and serves as the basis for the YOLO algorithm. As part of the transfer learning, YOLOv4 training started with pre-trained weights that were originally trained on the MS-COCO (Microsoft Common Objects in Context) dataset [36], which contains a wide range of 80 object classes. Training was conducted on the training set, while evaluation was performed on the validation set. When the mean average precision of the model evaluated on the validation set did not increase, the training was stopped, as seen in Figure 4. The best weights with the highest mean average precision were taken for the designated weed detection model.

#### 2.5.2. EfficentDet

EfficientDet is a real-time object detection model written in Tensorflow [37] and Keras [38] that utilizes a weighted bi-directional feature pyramid network (BiFPN) to learn input features while incorporating multi-scale feature fusing for box/class prediction.

A pre-trained model (EfficientDet D0 512 × 512) from a collection of models pre-trained on the COCO 2017 dataset provided by Tensorflow 2 Detection Model Zoo [39] served as the starting point for training the EfficientDet weed detection model. The training was carried out while monitoring the validation loss (Figure 5), average precision (Figure 6), and recall (Figure 7) and stopped when the loss did not decrease and the precision and recall did not increase (around 30 K).

#### 2.5.3. CenterNet

CenterNet represents objects as a set of keypoints, reducing the need for anchor boxes and simplifying the process by predicting the bounding boxes directly.

The training of the CenterNet model utilized a pre-trained model (CenterNet Resnet101 V1 FPN 512 × 512) from Tensorflow 2 Detection Model Zoo, which was trained over the Resnet101 [40] backbone as the starting network. Total validation loss (Figure 8), precision (Figure 9), and recall (Figure 10) were monitored during the training. Table 1 shows the architecture differences between the models used in this study.

### 2.6. Platforms

The weed detection models were trained on a deep-learning-capable computer equipped with a 32-core Intel I9 CPU (Intel, Santa Clara, CA, USA, Nvidia RTX 2080 Ti GPUs (4352-CUDA cores), and 128 GB RAM. Inference speed and accuracy were compared between the deep learning computer and an artificial intelligence (AI)-embedded computer designed specifically for autonomous machines, an Nvidia Jetson Xavier AGX (Figure 11) equipped with an 8-core NVIDIA Carmel Arm^®^v8.2 64-bit CPU 8 MB L2 + 4 MB L3, 512-core NVIDIA Volta architecture GPU with 64 Tensor Cores, and 32 GB of RAM.

### 2.7. Evaluation Metrics

As the focus lay on the inference of the detection models, several metrics were compared when running on the two platforms. Model accuracy metrics such as precision and recall, which were summarized by the average precision (AP) value and mean average precision (mAP) evaluated under different Intersection-over-Union (IoU) thresholds, and speed metrics such as inference time and frames per second (fps) were considered.

Precision measures how well the positive predictions match the ground truth.
$$Precision=\frac{True\;positives}{True\;positives+False\;positives}$$

Recall measures how many relevant predictions are made out of all predictions.
$$Recall=\frac{True\;positives}{True\;positives+False\;negatives}$$

The average precision (AP) represents the weighted average of all precision values at each precision–recall curve threshold, where the weight is the increase in recall. This value summarizes the precision–recall curve into a single value.
$$AP=\sum_{k=0}^{k=n-1}\left[Recalls\left(k\right)-Recalls\left(k+1\right)\right]\times Precisions\left(k\right)\;Recalls\left(n\right)=0,\;Precisions\left(n\right)=1,\;n=Number\;of\;thres\;holds$$

The Intersection over Union (IoU) indicates the overlap of the predicted bounding box coordinates with the ground-truth box [41], as shown in Figure 12. When the predicted bounding box closely resembles the ground-truth box, the IoU is higher. In deep learning object detection models, multiple bounding boxes are predicted for objects, but only those with an IoU higher than a certain threshold are considered as positively predicted boxes.
$$IoU=\frac{Area\;of\;Overlap}{Area\;of\;Union}$$

The Mean average precision (mAP) represents the average of the weighted means of precision at each IoU threshold. It is calculated by averaging the average precision (AP) for each class across a number of classes.
$$mAP=\frac{1}{N}\sum_{i=1}^{N}{\left(AP\right)}_{i}$$

The inference time refers to the time it takes for a model to make a prediction on a single image, while the number of frames per second (fps) indicates the frequency at which inference is performed on consecutive images in a video stream. For real-time applications, these are crucial metrics because an excessive inference delay can lead to the machine being unable to respond in time. The inference time was calculated by running the models on a set of weed images and averaging the time over the number of images. On the other hand, fps was obtained by running the models on weed videos while recording the reciprocal of execution time for each frame. These two metrics varied among the models as well as platforms, while the other metrics only varied among the models.

### 2.8. Mobile Optimized Solution

The prediction speed is a critical aspect of a real-time detection system, and due to the fact that in real scenarios the embedded computer runs other applications for robot control in addition to the detection program, the inference speed may be impacted further. Other variants of deep learning models optimized for speed have been developed by sacrificing some precision through reducing the neural network size. YOLOv4 has a lightweight compressed version, YOLOv4-tiny, with a simpler network structure and reduced parameters to make it ideal for mobile and embedded devices. YOLOv4-tiny can be used for faster training and inference than YOLOv4; however, its accuracy suffers. YOLOv4 was also compared to its lighter version YOLOv4-tiny in terms of its viability for weed detection on the embedded platform.

## 3. Results

### 3.1. Model Comparisons

The results, as shown in Table 2, obtained when the models were evaluated on 600 labeled test images from the dataset (20% of the dataset) using COCO metrics, indicate that EfficientDet and CenterNet had similar performance, with an overall mean average precision of 71.3% and 70.6%, respectively, which was better than YOLO, with an mAP of 61.6% at IoU = 0.5–0.95. This mAP was calculated by taking the average mAP over IoU thresholds ranging from 0.5 to 0.95 with a step size of 0.05. At IoU = 0.5, EfficientDet outperformed all the other models (97.4%). However, the other models achieved satisfactory scores of 93.8% (CenterNet) and 93.4% (YOLOv4). Overall, EfficientDet had a slight edge over the other models in terms of accuracy metrics.

Visual observation showed no significant difference in detection, except for a few images where EfficientDet had better predictions; for example, in Figure 13 and Figure 14, YOLOv4 and CenterNet failed to detect crowfoot grass, but EfficientDet detected it in Figure 15.

Regarding their performance in detecting individual weed species, the models presented good results for class evaluation using PASCAL VOC mAP @ IoU = 0.5 metrics, except for CenterNet and EfficientDet when detecting purple nutsedge—they achieved mAP@0.5 scores of 0.06% and 0.07%, respectively, as shown in Table 3. This could be attributed to the limited number of training images for purple nutsedge and the similarity between yellow and purple nutsedge. However, YOLOv4 performed significantly better on purple nutsedge, with an mAP of 79.4% at IoU = 0.5. This difference is evident even in the visual inspections in Figure 16 and Figure 17, where the Centernet and EfficientDet models detected only one of two purple nutsedge plants, but YOLOv4 in Figure 18 successfully detected both weeds.

#### Confusion Matrices

The confusion matrices for the YOLOv4, EfficientDet, and CenterNet models on the test dataset are shown in Figure 19, Figure 20, and Figure 21, respectively. EfficientDet performed the best, achieving accuracies of 95% or higher for 9 out of 13 weed classes, while YOLOv4 and CenterNet achieved accuracies of 95% or higher for 7 out of 13 weed classes. However, YOLOv4 was less accurate in more weed classes than CenterNet; for example, YOLOv4 identified crowfoot grass accurately 76% of the time, while CenterNet identified crowfoot grass 83% of the time.

### 3.2. Inference Time

For real-time robotic applications, the speed of detection is crucial. When the models were run on 600 images on both the deep learning PC and embedded computer, YOLOv4 performed significantly better on both platforms, as shown in Table 4, with an average of 18 ms per image on the PC and, importantly, just 80 ms per image on the embedded computer. CenterNet performed better than EfficientDet on the PC, predicting at 44 ms per image versus 66 ms; however, EfficientDet outperformed CenterNet on the embedded computer, achieving 102 ms versus 140 ms.

### 3.3. Frames Per Second (fps)

When the models were run on a video with a resolution 1280 × 720 on both platforms, YOLOv4 outperformed the other models, as shown in Table 5, achieving 51 fps on the PC and 14 fps on the embedded computer, while EfficientDet performed better than CenterNet on the embedded computer, reaching 12 fps versus 8 fps.

### 3.4. Improvement with YOLOv4-Tiny

Regarding the balance of accuracy and speed, YOLOv4 performed better than the other models on the embedded computer. An object detection speed of 14 fps may suffice for many real-time applications; however, certain applications demand a higher speed. An ideal model can identify and locate objects in real time with rapid inference while maintaining a baseline level of accuracy. To assess this, the YOLOv4-tiny model was trained on the same dataset as YOLOv4. During accuracy testing, YOLOv4-tiny achieved a precision of 81%, recall of 88%, and reduced mAP at IoU = 0.5 of 89.2%, as shown in Table 6. Impressively, when tested on the embedded computer, it achieved an inference time of 24.5 ms and 52 frames per second, surpassing YOLOv4.

## 4. Discussion

Real-time weed detection is crucial for precision mapping and the removal of weeds in agricultural fields. To achieve effective precision weed removal, robotic platforms are commonly employed. As these platforms often use embedded computers for their portability, it becomes important to evaluate the performance of various weed detection models on these embedded systems and identify the ideal model for real-time weed detection. Despite the prevalence of research in weed detection, there has been limited testing of these solutions on embedded computers to assess their practicality. Our approach involved comparing the performance of three real-time deep learning models—YOLOv4, EfficientDet, and CenterNet—in detecting 13 different species of weeds. This comparison focused on the accuracy of the models and their inference speed.

Our weed dataset was meticulously curated, encompassing images captured under various weather conditions and at different times of the day, growth stages, and camera angles. Additionally, data augmentation was employed to enhance the diversity of the training samples, following methodologies outlined in studies such as [42,43].

Each of the three deep learning models achieved a mean average precision greater than 93% at a 50% Intersection over Union (IoU) threshold. The models YOLOv4, EfficientDet, and CenterNet exhibited COCO mean average precision values of 61.6%, 71.3%, and 70.6%, respectively. In terms of inference times, the models performed at 18 ms, 66 ms, and 44 ms on a deep learning computer and at 80 ms, 102 ms, and 140 ms on an embedded computer, respectively.

Comparing our results to other weed detection studies, refs. [14] and [15] achieved accuracy levels exceeding 87% and 91%, respectively, in weed classification. However, these studies were conducted in controlled environments, and there was no indication of inference speed or tests on embedded computers. Conversely, solutions using computer vision algorithms like [16,17,18] achieved over 90% accuracy in discriminating weeds from plants. However, these solutions classified only a few weed species compared to our 13 species, and there was no indication of inference speed to evaluate their real-time capabilities.

When considering similar solutions utilizing deep learning, the accuracies align closely with our observations. For example, ref. [22] achieved average precision values ranging from 75% for the VGG16 network to 97% using the ResNet-50 and Xception networks on 12 different plant species. Another comparable deep learning method [27], evaluating the performance of 35 models on 15 weed classes, achieved accuracies from 50% for the low-performing model MnasNet to 98% for the top-performing ResNext101 model. However, their reported inference times ranging from 188 ms to 338 ms were slower than our models’ inference times.

Considering practical robot usage in the field, embedded computers are preferred. The authors of [19] attempted to evaluate the performance of segmenting weeds using customized MobileNet and DenseNet networks on an embedded computer (Raspberry Pi). The solution achieved an inference time of 50 ms to 100 ms. Although this inference time was impressively shorter than our best inference time on an embedded computer obtained through YOLOv4 (80 ms), it is noteworthy that our recommended solution for embedded systems, YOLOv4-tiny, boasts the best inference time of 24.5 ms. Future studies should evaluate the real-time performance on a robotic platform in an agricultural field.

## 5. Conclusions

Three deep learning models—YOLOv4, EfficientDet, and CenterNet—were trained and tested for their effectiveness in detecting thirteen different species of weed using two platforms: a deep-learning-capable computer and an embedded computer. The experiment aimed to assess their suitability for real-time robotic applications. It was observed that, with a mean average precision of 93.4% at an IoU threshold of 50%, an inference speed of 80 ms, and 14 fps on an embedded computer, YOLOv4 is better suited for real-time robotic applications due to its balanced performance between accuracy and inference speed. Furthermore, recognizing that some real-time robotic applications require a higher speed without compromising the accuracy too much, a lightweight version of YOLOv4, YOLOv4-tiny, was trained and tested in an embedded system. Despite its smaller size, YOLOv4-tiny impressively achieved a mean average precision of 89% at a 50% IoU threshold, which is approximately 4.7% less precise than YOLOv4. The model performed inference very rapidly on an embedded computer, with a speed of 24.5 ms and 52 fps.

Due to its speed of detection in an embedded system and its satisfactory accuracy, YOLOv4-tiny is recommended for real-time robotic applications that involve weed detection.

## Figures and Tables

**Figure 1 sensors-24-00514-f001:**
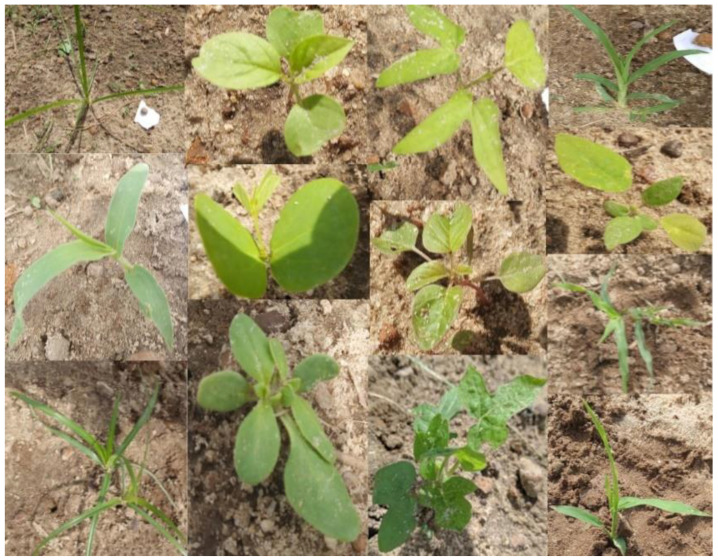
Examples of images of 13 weed species.

**Figure 2 sensors-24-00514-f002:**
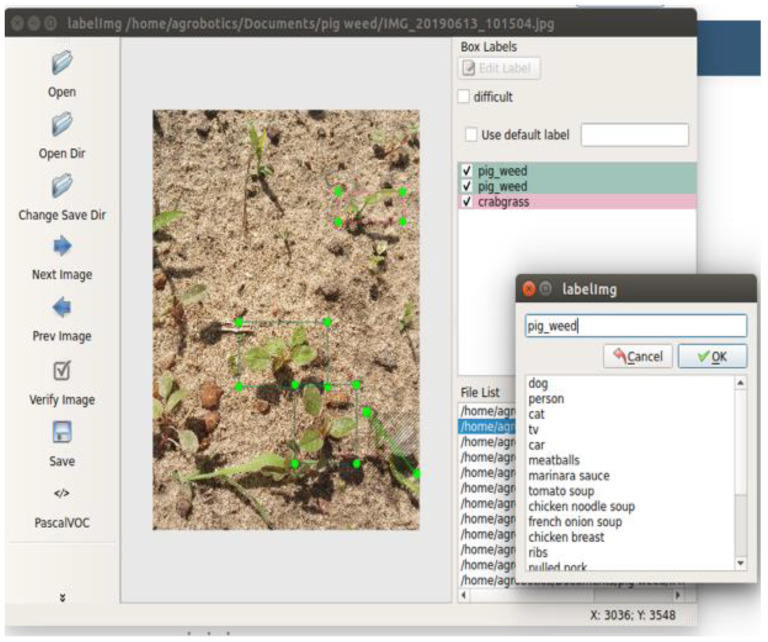
Example of labeling weed image using LabelImg.

**Figure 3 sensors-24-00514-f003:**
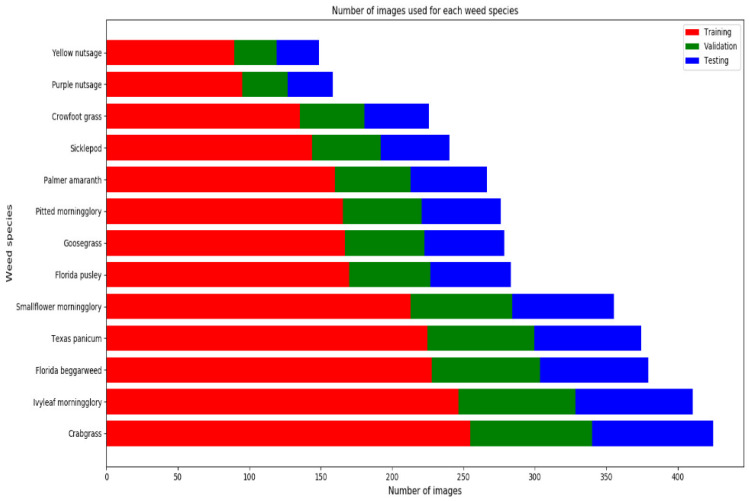
Labeled dataset split.

**Figure 4 sensors-24-00514-f004:**
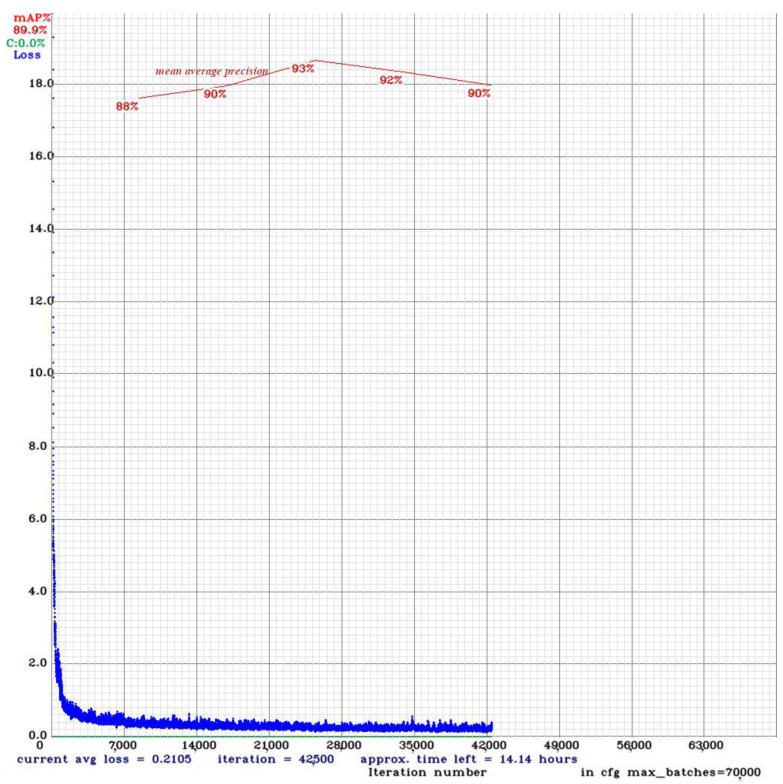
YOLOv4 training on darknet platform.

**Figure 5 sensors-24-00514-f005:**
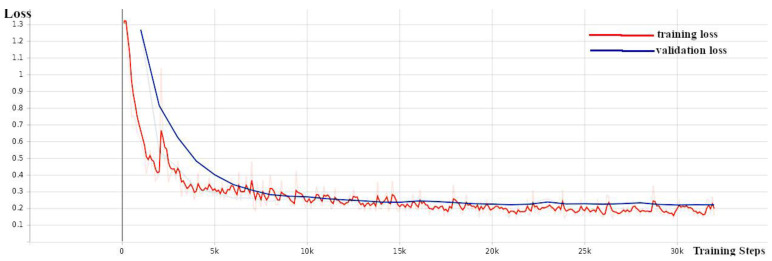
EfficientDet: total loss against number of training steps—training loss (orange) and validation loss (blue).

**Figure 6 sensors-24-00514-f006:**
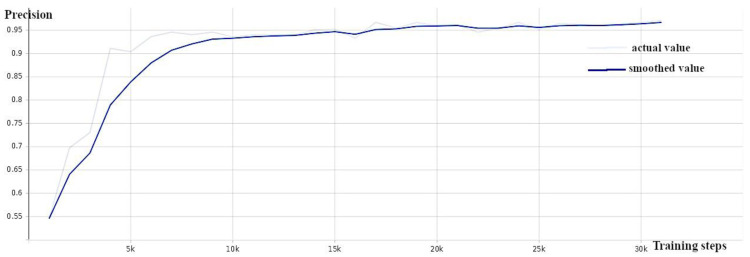
EfficientDet: precision (mAP@0.5) against number of training steps.

**Figure 7 sensors-24-00514-f007:**
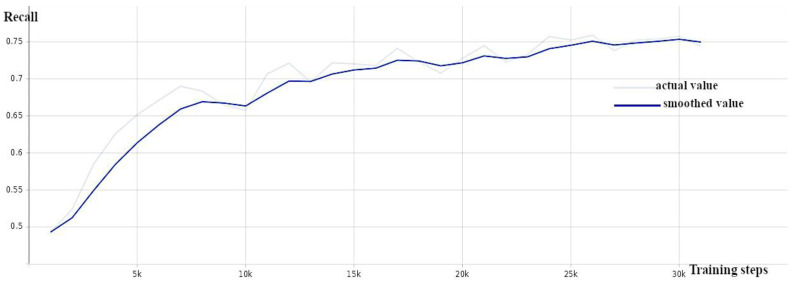
EfficientDet: recall against number of training steps.

**Figure 8 sensors-24-00514-f008:**
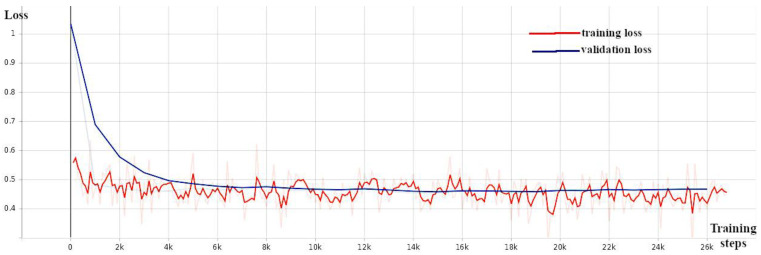
CenterNet: total loss against number of training steps—training loss (orange) and validation loss (blue).

**Figure 9 sensors-24-00514-f009:**
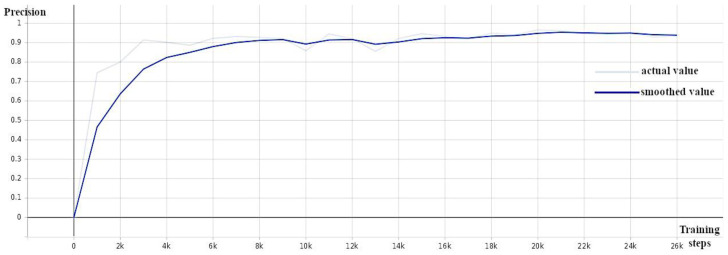
CenterNet: precision (mAP@0.5) against number of training steps.

**Figure 10 sensors-24-00514-f010:**
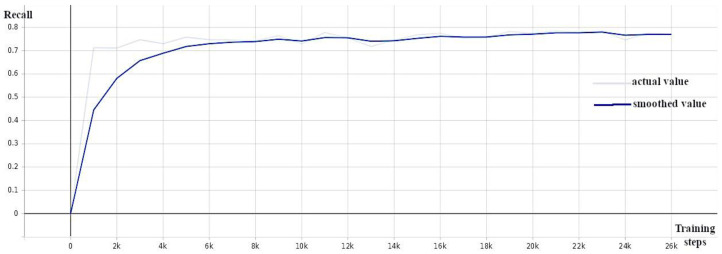
CenterNet: recall against number of training steps.

**Figure 11 sensors-24-00514-f011:**
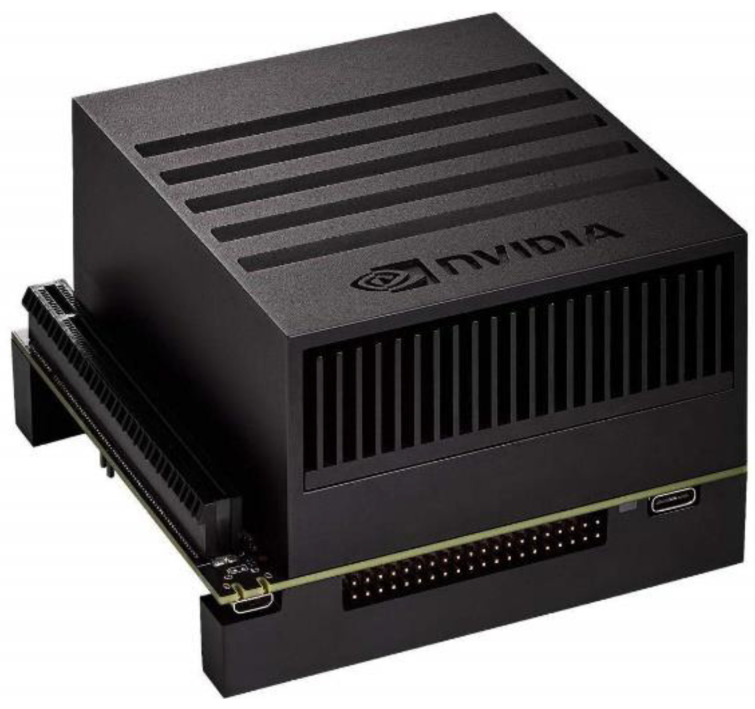
Nvidia Jetson Xavier AGX.

**Figure 12 sensors-24-00514-f012:**
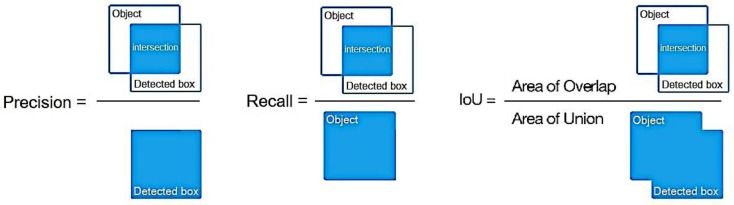
Precision, recall, and IoU illustration.

**Figure 13 sensors-24-00514-f013:**
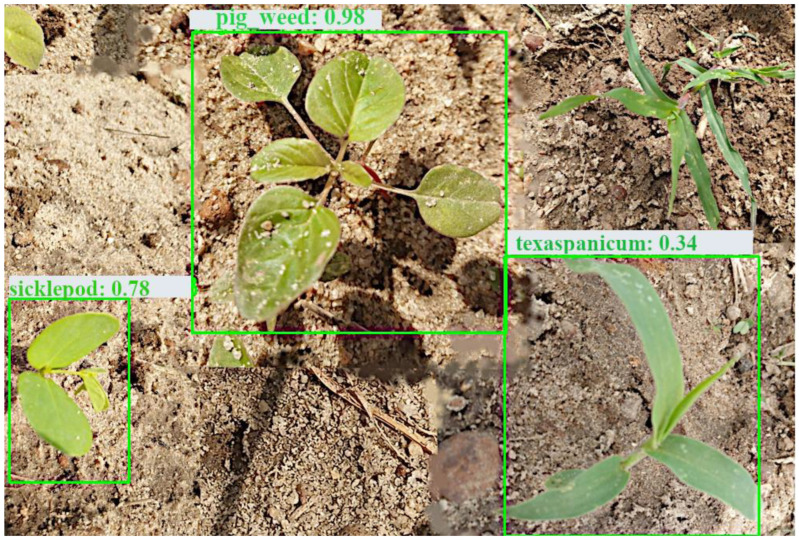
YOLOv4 detection—detected 3 weeds.

**Figure 14 sensors-24-00514-f014:**
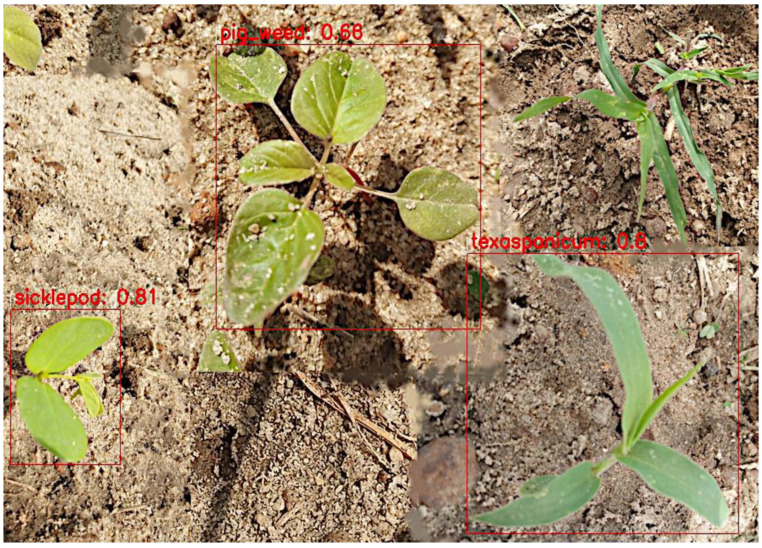
CenterNet detection—detected 3 weeds.

**Figure 15 sensors-24-00514-f015:**
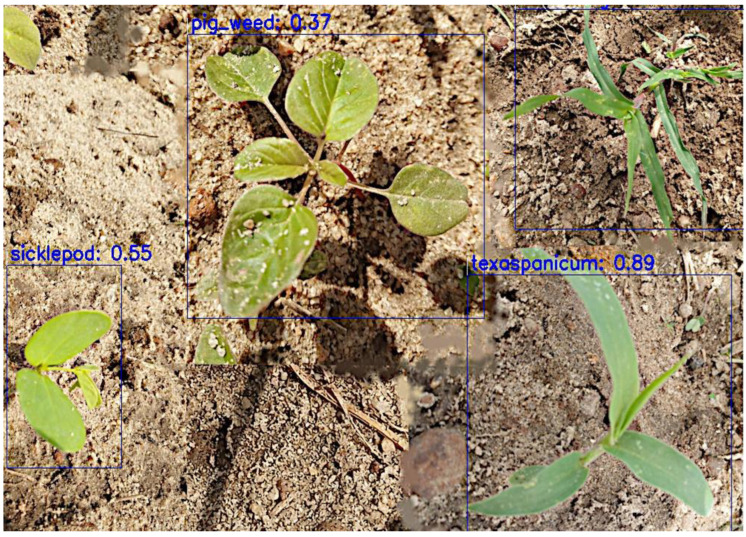
EfficientDet detection—detected all 4 weeds.

**Figure 16 sensors-24-00514-f016:**
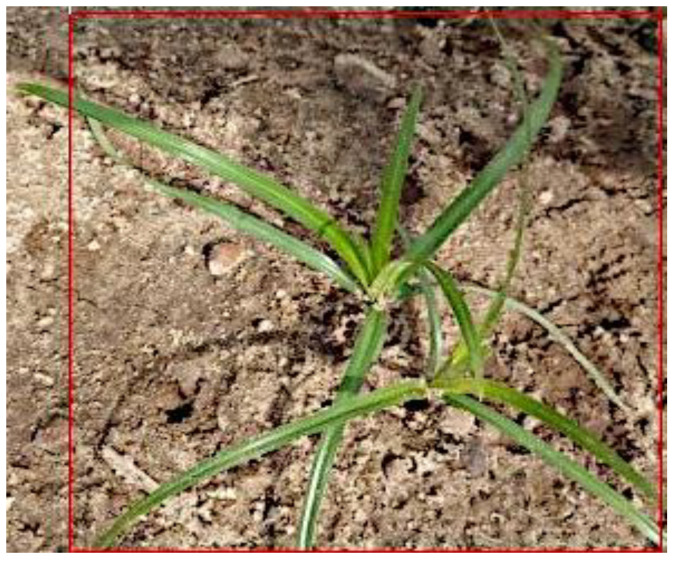
Single purple nutsedge plant detected by CenterNet.

**Figure 17 sensors-24-00514-f017:**
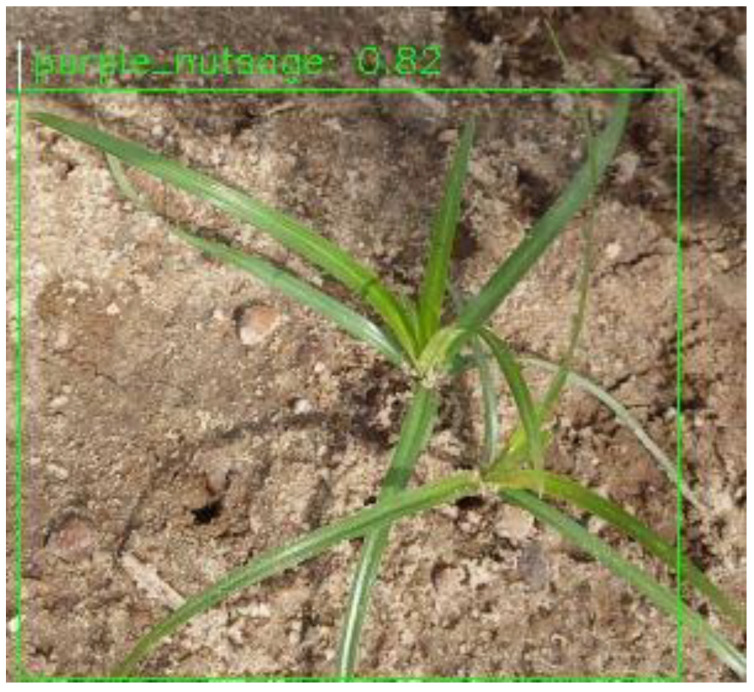
Single purple nutsedge plant detected by EfficientDet.

**Figure 18 sensors-24-00514-f018:**
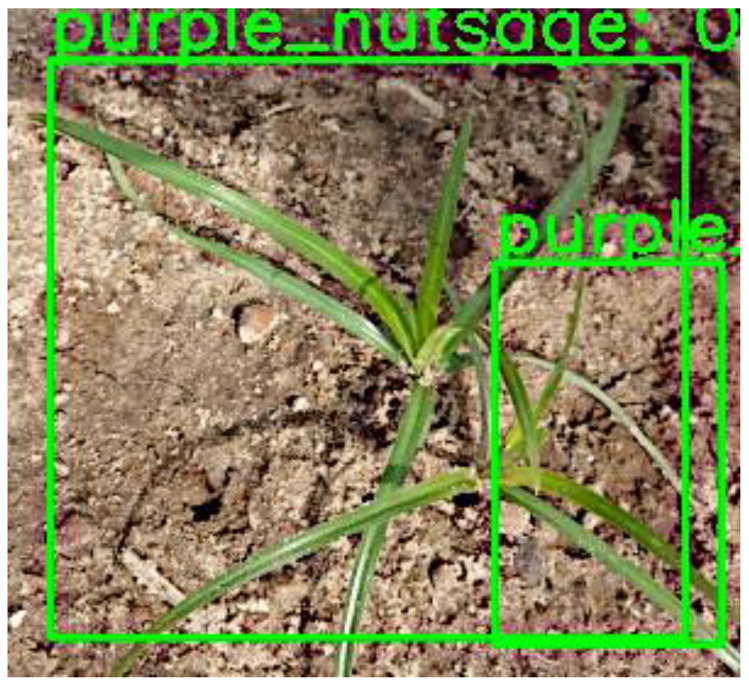
Two purple nutsedge plants detected by YOLOv4.

**Figure 19 sensors-24-00514-f019:**
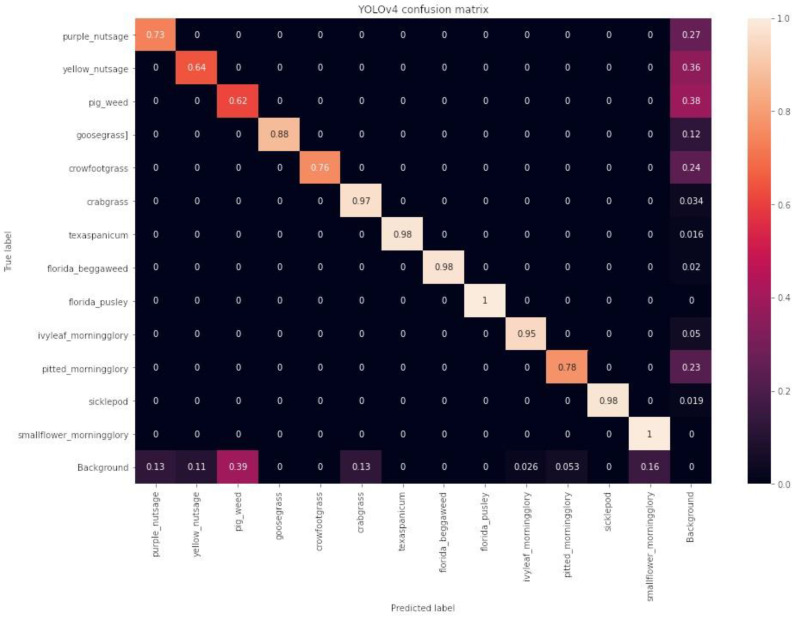
YOLOv4 confusion matrix on the test dataset.

**Figure 20 sensors-24-00514-f020:**
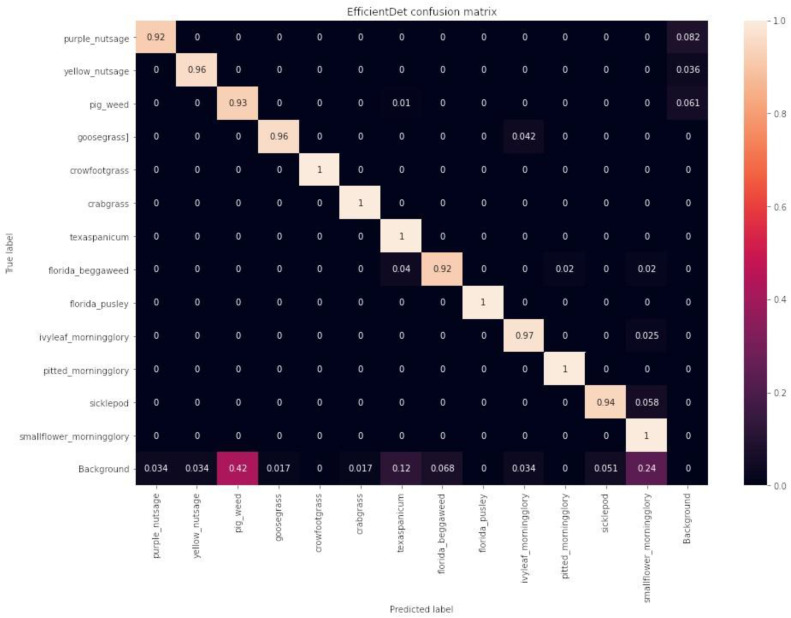
EfficientDet confusion matrix on the test dataset.

**Figure 21 sensors-24-00514-f021:**
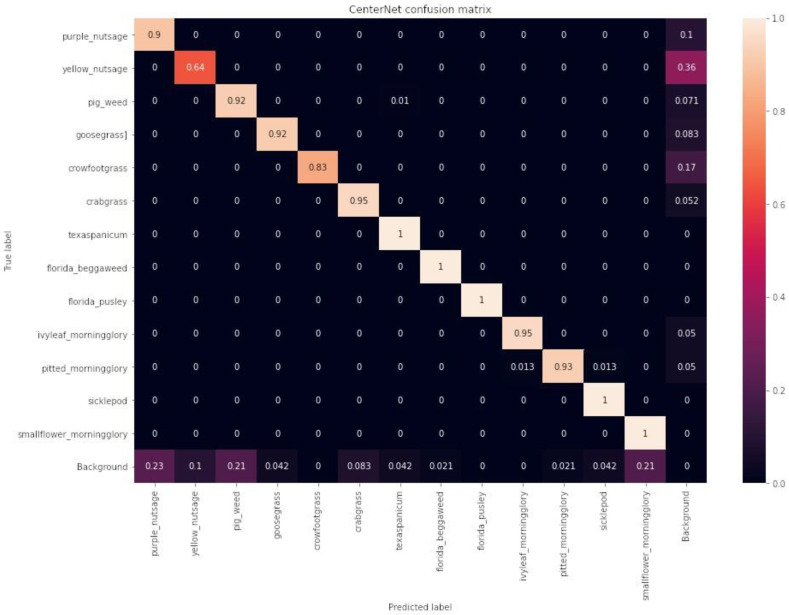
CenterNet confusion matrix on the test dataset.

**Table 1 sensors-24-00514-t001:** Architecture comparisons for the models used in this study.

	YOLOv4	EfficientDet	CenterNet
Number of stages	One-stage	One-stage	One-stage
Backbone	CSPDarknet53	EfficientNetB0	Resnet101
Number of layers	53	237	101
Object detection method	Anchor-based	Anchor-based	Anchor-free
Input size	416 × 416	512 × 512	512 × 512

**Table 2 sensors-24-00514-t002:** Model accuracy performance comparisons.

Metric	YOLOv4	CenterNet	EfficientDet
mAP@ IoU = 0.5–0.95	0.616	0.706	0.713
mAP@ IoU = 0.5	0.934	0.938	0.974
mAP@ IoU = 0.75	0.703	0.809	0.819
Average recall	0.660	0.714	0.708

**Table 3 sensors-24-00514-t003:** Model mAP@0.5 for individual weed classes.

Weed Species	YOLOv4	CenterNet	EfficientDet
Smallflower morningglory	0.994	0.998	0.990
Sicklepod	1.000	0.998	1.000
Pitted morningglory	0.899	0.990	1.000
Ivyleaf morningglory	0.998	0.987	1.000
Florida pusley	1.000	1.000	1.000
Florida beggarweed	1.000	1.000	0.998
Texas panicum	0.999	0.999	1.000
Crabgrass	0.997	0.991	0.999
Crowfoot grass	0.808	0.957	1.000
Goosegrass	1.000	0.967	1.000
Palmer amaranth	0.712	0.911	0.926
Yellow nutsedge	0.763	0.653	0.940
Purple nutsedge	0.794	0.0006	0.0007

**Table 4 sensors-24-00514-t004:** Inference time (ms).

Platform	YOLOv4	CenterNet	EfficientDet
Deep learning computer	18	44	66
Jetson Xavier AGX	80	140	102

**Table 5 sensors-24-00514-t005:** Number of frames per second achieved.

Platform	YOLOv4	CenterNet	EfficientDet
Deep learning computer	51	40	22
Jetson Xavier AGX	14	8	12

**Table 6 sensors-24-00514-t006:** YOLOv4-tiny evaluation results compared to YOLOv4 on embedded computer.

Metric	YOLOv4-Tiny	YOLOv4
Precision	0.81	0.95
Recall	0.88	0.89
mAP @ IoU = 0.5	0.89	0.934
Inference on Jetson Xavier AGX (ms)	24.5	80
FPS on Jetson Xavier AGX	52	14

## Data Availability

The data presented in this study are available on request from the corresponding author.

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
