# Peer review of "Evaluation of Inference Performance of Deep Learning Models for Real-Time Weed Detection in an Embedded Computer"

_sensors, 2024, doi:10.3390/s24020514_

Round 1
Reviewer 1 Report
Comments and Suggestions for Authors
-
-
What is the primary motivation for precision weed control in agricultural fields, and how does it relate to the development of autonomous machines for weed detection and removal?
-
Why is there a need for alternatives to herbicides in weed detection, and what challenges are posed by the resistance of weed species to herbicides?
-
How have advances in deep learning technology improved the robustness of weed detection tasks, and what computational resources are typically required for deep learning algorithms?
-
What is the significance of developing deep learning models that are computationally lightweight for deployment in robotic platforms with embedded computers?
-
Which three real-time capable deep learning models are evaluated in the paper for weed detection, and what is the specific embedded computer used for testing their performance?
-
What criteria are used to assess the weed detection performance of the models, and how many species of weeds are considered in the evaluation?
-
What were the results regarding the weed detection performance of YOLOv4, EfficientDet, and CenterNet, particularly in terms of inference speed and mean average precision?
-
Why is YOLOv4 considered a better-performing model in the context of real-time weed detection, and what were its specific performance metrics when run on an embedded computer?
-
Considering the need for even greater speed in some real-world applications, what alternative model is tested for improved performance, and how does it compare to YOLOv4 in terms of inference speed and mean average precision?
-
The discussion section needs to be strengthened and supported with literature
-
The conclusion section is mandatory.
-
The following papers can be referred to for algorithms (a)Target-driven visual navigation in indoor scenes using reinforcement learning and imitation learning (b) An attention-based cascade R-CNN model for sternum fracture detection in X-ray images (c) A robust deformed convolutional neural network (CNN) for image denoising
-
A confusion matrix needs to be added to compute the performance metrics.
-
Moderate editing of the English language required
Author Response
Thank you very much for taking the time to review this manuscript. Please find the detailed responses below and the corresponding revisions/corrections highlighted/in track changes in the re-submitted files.
- Comment 1: What is the primary motivation for precision weed control in agricultural fields, and how does it relate to the development of autonomous machines for weed detection and removal?
Response: The primary motivation for precision weed control in agricultural fields is to increase efficiency in control, reduce costs, minimize the use of herbicide, and minimize the labor-intensive nature of manual and mechanical weeding. Development of autonomous machines for weed detection and removal directly supports the primary motivation
- Comment 2: Why is there a need for alternatives to herbicides in weed detection, and what challenges are posed by the resistance of weed species to herbicides?
Response: The need for herbicides alternatives is necessitated by the evolution of herbicide-resistant weed populations and its negative impact on the environment. Resistance of weed species to herbicides renders the herbicide weed controlling method ineffective
- Comment 3: How have advances in deep learning technology improved the robustness of weed detection tasks, and what computational resources are typically required for deep learning algorithms?
Response: Deep learning technologies utilize examples from different scenarios and scenery (lighting conditions, shape, size, color, growth stages, viewing angle, etc.) through training to improve the robustness of detection. Typically, deep learning would require computers with powerful GPUs to handle processing the massive data from images.
- Comment 4: What is the significance of developing deep learning models that are computationally lightweight for deployment in robotic platforms with embedded computers?
Response: Most robotic platforms employ embedded computers due to their portability and reduced power consumption, however, these computers have limited computational capacity. It is imperative to develop deep learning models that are lightweight enough to run in embedded computers for autonomous robotic platforms to run effectively.
- Comment 5: Which three real-time capable deep learning models are evaluated in the paper for weed detection, and what is the specific embedded computer used for testing their performance?
Response: YOLOv4, EfficientDet, and CenterNet deep learning models were evaluated in the study, and the embedded computer used was Nvidia Jetson Xavier AGX.
- Comment 6: What criteria are used to assess the weed detection performance of the models, and how many species of weeds are considered in the evaluation?
Response: Detection accuracy of the models (Precision, recall) and inference speed are the criteria used to evaluate the detection performance. 13 weed species were considered.
- Comment 7: What were the results regarding the weed detection performance of YOLOv4, EfficientDet, and CenterNet, particularly in terms of inference speed and mean average precision?
Response: Overall YOLOv4 performed better in an embedded computer (mean average precision ([email protected]) of 93.4% and inference speed of 80 ms), than EfficientDet (mAP of 97.4% and inference speed of 102 ms) and CenterNet (mAP of 93.8% and inference speed of 140 ms)
- Comment 8: Why is YOLOv4 considered a better-performing model in the context of real-time weed detection, and what were its specific performance metrics when run on an embedded computer?
Response: YOLOv4 was considered better-performing in embedded computer specifically because of its low inference speed of 80 ms, high frames processed per second (14) in embedded computer, while having a decent mean average precision of 93.4% (mAP @ 0.5)
- Comment 9: Considering the need for even greater speed in some real-world applications, what alternative model is tested for improved performance, and how does it compare to YOLOv4 in terms of inference speed and mean average precision?
Response: YOLOv4-tiny was preferred as an alternative for even greater speed for practical real-time applications because of its impressive inference speed of 24.5 ms in the embedded computer compared to YOLOv4’s of 80 ms despite being only about 4.7% less precise than YOLOv4.
- Comment 10: The discussion section needs to be strengthened and supported with literature
Response: Thank you for the comment. We have added a Discussion section supported with literature
- Comment 11: The conclusion section is mandatory.
Response: Thank you for pointing this out. We added a conclusion section
- Comment 12: The following papers can be referred to for algorithms (a)Target-driven visual navigation in indoor scenes using reinforcement learning and imitation learning (b) An attention-based cascade R-CNN model for sternum fracture detection in X-ray images (c) A robust deformed convolutional neural network (CNN) for image denoising
Response: Thank you for the suggestion. We have cited (b) which is more relevant to the study.
- Comment 13: A confusion matrix needs to be added to compute the performance metrics.
Response: Thank you for pointing this out. We added confusion matrices for the three models on the test dataset in Results section.
Reviewer 2 Report
Comments and Suggestions for Authors
General comments
This study examines the utilization of deep learning models for the purpose of precise weed control. This study is driven by the necessity for efficient alternatives to herbicides, as their effectiveness is diminishing due to the emergence of weed species that are resistant to them.
The study investigates the application of three advanced deep learning models, namely YOLOv4, EfficientDet, and CenterNet, on an embedded computer system (Nvidia Jetson Xavier AGX) to achieve real-time weed detection. The authors strive to achieve a harmonious equilibrium between computational efficiency and detection precision, ensuring that the system is well-suited for deployment in robotic platforms such as ground rovers and UAVs.
Main achievements of authors:
1. The performance of YOLOv4 surpasses that of the other models, with an average inference speed of 80ms per image and a mean average precision of 93.4% at a 50% Intersection over Union (IoU) threshold. For situations that demand higher velocity, a more lightweight variant, YOLOv4-tiny, achieves an average inference speed of 24.5ms per image with a little lower accuracy of 89%.
2. Model comparison: The study conducts a comprehensive examination of the models, comparing them based on several parameters such as precision, recall, average precision, inference time, and frames per second.
3. Data collection and processing: A total of 5000 pictures depicting 13 distinct weed species were gathered and subjected to processing. The dataset was expanded by data augmentation, and the models were trained utilizing transfer learning techniques.
Suggestions for authors to consider in order to enhance the caliber of their paper:
In order to further improve the quality and effectiveness of the article titled "Evaluation of Inference Performance of Deep Learning Models for Real-Time Weed Detection in an Embedded Computer," the following suggestions can be taken into account:
1. The research adequately examines the application of deep learning models in weed identification. However, it would be advantageous to provide a more comprehensive examination of the environmental and economic consequences associated with weed management techniques. Providing a comparison with conventional herbicide-based methods would enhance the overall comprehension of the research's importance.
2. Enhanced methodological details: Elaborating on the dataset could provide useful insights, including precise information on the picture acquisition technique, the conditions under which the images were obtained, and any preprocessing procedures that were employed. This would facilitate the reproduction and verification of the study's results.
3. The paper should be enhanced by conducting robustness testing, which involves additional testing in diverse environmental variables such as varying lighting, weather conditions, and phases of weed growth. This would showcase the resilience and pragmatic usefulness of the models in actual agricultural environments.
4. An examination of energy usage, namely power consumption and battery life consequences, would be advantageous for deploying these models on embedded devices. This characteristic is vital for agricultural robotics applications that require real-time capabilities.
5. Proposing user-friendly interfaces for end-users such as farmers and agronomists to interact with the technology could help bridge the divide between technical feasibility and practical applicability.
Important to verify
Page 10 – Line 222 – where is the reference?
The work exhibits a well-organized structure, commencing with a thorough introduction to the issue of weed detection and control. It proceeds with a meticulous methodology section that elucidates the process of data collecting, labeling, and model training. Subsequently, the research presents a complete evaluation of the models employed. The discussion and conclusion sections emphasize the practical consequences of the study, specifically highlighting the potential of YOLOv4 and its small form for real-time weed detection in agricultural robotics.
Overall, the research makes a substantial contribution to the field of precision agriculture by assessing the practicality of implementing deep learning models on embedded devices to identify and classify weeds. The analysis provides significant perspectives on the efficacy of various models and makes a compelling argument for the utilization of YOLOv4 and YOLOv4-tiny in real-time applications. This study's comprehensive approach, along with its meticulous experimental results, establishes it as a significant resource for scholars and practitioners in the agricultural robotics sector.

moderate checking of English language is required
Author Response
Thank you very much for taking the time to review this manuscript. Please find the detailed responses below and the corresponding revisions/corrections highlighted/in track changes in the re-submitted files.
- Comment 1: The research adequately examines the application of deep learning models in weed identification. However, it would be advantageous to provide a more comprehensive examination of the environmental and economic consequences associated with weed management techniques. Providing a comparison with conventional herbicide-based methods would enhance the overall comprehension of the research's importance.
Response: Thank you for this suggestion. The comparisons of weed management techniques were out of scope for this study, however, the study is part of a bigger project to develop autonomous weeding robotic systems. So future studies will compare their impact with herbicide-based methods.
- Comment 2: Enhanced methodological details: Elaborating on the dataset could provide useful insights, including precise information on the picture acquisition technique, the conditions under which the images were obtained, and any preprocessing procedures that were employed. This would facilitate the reproduction and verification of the study's results
Response: Agree. We have added more elaboration of the dataset collection in Material and Methods (2.1) and Discussion sections. The dataset was collected at different times of the day, weather conditions, growth stages, and camera angles. The training data was also augmented through different image manipulation techniques to increase the dataset (2.4)
- Comment 3: The paper should be enhanced by conducting robustness testing, which involves additional testing in diverse environmental variables such as varying lighting, weather conditions, and phases of weed growth. This would showcase the resilience and pragmatic usefulness of the models in actual agricultural environments
Response: Thank you for the comment. The dataset was split into training, validation, and testing. The testing dataset encompassed diverse images obtained from different weather conditions, times of the day, and growth stages (at early stages of growth)
- Comment 4: An examination of energy usage, namely power consumption and battery life consequences, would be advantageous for deploying these models on embedded devices. This characteristic is vital for agricultural robotics applications that require real-time capabilities.
Response: Thank you for the suggestion. It would have been interesting to explore this aspect. However, for this study we aimed to evaluate and find appropriate models that can be used in embedded computers which are common for robotic application. As part of a bigger robotic project, we will incorporate this aspect in our future studies.
- Comment 5: Proposing user-friendly interfaces for end-users such as farmers and agronomists to interact with the technology could help bridge the divide between technical feasibility and practical applicability.
Response: Thank you for this suggestion. This is a very important part of the project since the system would be useless if the intended users were not able to interact with it. We always priorities user-friendliness and simplification aspects when we design our system interfaces.
- Comment 6: Page 10 – Line 222 – where is the reference?
Response: Thank you for noting. Reference is added.
Round 2
Reviewer 1 Report
Comments and Suggestions for Authors
Congrats to the authors.
Comments on the Quality of English LanguageMinor editing of the English language is required